# TernGrad: Ternary Gradients to Reduce Communication in Distributed Deep Learning

**Wei Wen[1], Cong Xu[2], Feng Yan[3], Chunpeng Wu[1], Yandan Wang[4], Yiran Chen[1], Hai Li[1]**

[1]Duke University, [2]Hewlett Packard Labs, [3]University of Nevada – Reno, [4]University of Pittsburgh
[1]{wei.wen, chunpeng.wu, yiran.chen, hai.li}@duke.edu
[2]cong.xu@hpe.com, [3]fyan@unr.edu, [4]yaw46@pitt.edu

## Abstract

High network communication cost for synchronizing gradients and parameters is the well-known bottleneck of distributed training. In this work, we propose *TernGrad* that uses ternary gradients to accelerate distributed deep learning in data parallelism. Our approach requires only three numerical levels $\{-1, 0, 1\}$, which can aggressively reduce the communication time. We mathematically prove the convergence of *TernGrad* under the assumption of a bound on gradients. Guided by the bound, we propose *layer-wise ternarizing* and *gradient clipping* to improve its convergence. Our experiments show that applying *TernGrad* on *AlexNet* doesn't incur any accuracy loss and can even improve accuracy. The accuracy loss of *GoogLeNet* induced by *TernGrad* is less than $2\%$ on average. Finally, a performance model is proposed to study the scalability of *TernGrad*. Experiments show significant speed gains for various deep neural networks. Our source code is available [1].

## 1  Introduction

The remarkable advances in deep learning is driven by data explosion and increase of model size. The training of large-scale models with huge amounts of data are often carried on distributed systems [1][2][3][4][5][6][7][8][9], where data parallelism is adopted to exploit the compute capability empowered by multiple workers [10]. *Stochastic Gradient Descent* (SGD) is usually selected as the optimization method because of its high computation efficiency. In realizing the data parallelism of SGD, model copies in computing workers are trained in parallel by applying different subsets of data. A centralized parameter server performs *gradient synchronization* by collecting all gradients and averaging them to update parameters. The updated parameters will be sent back to workers, that is, *parameter synchronization*. Increasing the number of workers helps to reduce the computation time dramatically. However, as the scale of distributed systems grows up, the extensive gradient and parameter synchronizations prolong the communication time and even amortize the savings of computation time [4][11][12]. A common approach to overcome such a network bottleneck is *asynchronous SGD* [1][4][7][12][13][14], which continues computation by using stale values without waiting for the completeness of synchronization. The inconsistency of parameters across computing workers, however, can degrade training accuracy and incur occasional divergence [15][16]. Moreover, its workload dynamics make the training nondeterministic and hard to debug.

From the perspective of inference acceleration, sparse and quantized *Deep Neural Networks* (DNNs) have been widely studied, such as [17][18][19][20][21][22][23][24][25]. However, these methods generally aggravate the training effort. Researches such as sparse logistic regression and Lasso optimization problems [4][12][26] took advantage of the sparsity inherent in models and achieved

remarkable speedup for distributed training. A more generic and important topic is how to accelerate the distributed training of dense models by utilizing sparsity and quantization techniques. For instance, Aji and Heafield [27] proposed to heuristically sparsify dense gradients by dropping off small values in order to reduce gradient communication. For the same purpose, quantizing gradients to low-precision values with smaller bit width has also been extensively studied [22][28][29][30].

Our work belongs to the category of gradient quantization, which is an orthogonal approach to sparsity methods. We propose *TernGrad* that quantizes gradients to ternary levels $\{-1, 0, 1\}$ to reduce the overhead of *gradient synchronization*. Furthermore, we propose *scaler sharing* and *parameter localization*, which can replace *parameter synchronization* with a low-precision gradient pulling. Comparing with previous works, our major contributions include: (1) we use ternary values for gradients to reduce communication; (2) we mathematically prove the convergence of *TernGrad* in general by proposing a statistical bound on gradients; (3) we propose *layer-wise ternarizing* and *gradient clipping* to move this bound closer toward the bound of standard SGD. These simple techniques successfully improve the convergence; (4) we build a performance model to evaluate the speed of training methods with compressed gradients, like *TernGrad*.

## 2 Related work

**Gradient sparsification.** Aji and Heafield [27] proposed a heuristic gradient sparsification method that truncated the smallest gradients and transmitted only the remaining large ones. The method greatly reduced the gradient communication and achieved 22% speed gain on 4 GPUs for a neural machine translation, without impacting the translation quality. An earlier study by Garg *et al.* [31] adopted the similar approach, but targeted at sparsity recovery instead of training acceleration. Our proposed *TernGrad* is orthogonal to these sparsity-based methods.

**Gradient quantization.** *DoReFa-Net* [22] derived from *AlexNet* reduced the bit widths of weights, activations and gradients to 1, 2 and 6, respectively. However, *DoReFa-Net* showed $9.8\%$ accuracy loss as it targeted at acceleration on single worker. S. Gupta *et al.* [30] successfully trained neural networks on MNIST and CIFAR-10 datasets using 16-bit numerical precision for an energy-efficient hardware accelerator. Our work, instead, tends to speedup the distributed training by decreasing the communicated gradients to three numerical levels $\{-1, 0, 1\}$. F. Seide *et al.* [28] applied 1-bit SGD to accelerate distributed training and empirically verified its effectiveness in speech applications. As the gradient quantization is conducted by columns, a floating-point scaler *per column* is required. So it cannot yield speed benefit on convolutional neural networks [29]. Moreover, "*cold start*" of the method [28] requires floating-point gradients to converge to a good initial point for the following 1-bit SGD. More importantly, it is unknown what conditions can guarantee its convergence. Comparably, our *TernGrad* can start the DNN training from scratch and we prove the conditions that promise the convergence of *TernGrad*. A. T. Suresh *et al.* [32] proposed stochastic rotated quantization of gradients, and reduced gradient precision to 4 bits for MNIST and CIFAR dataset. However, *TernGrad* achieves lower precision for larger dataset (*e.g.* ImageNet), and has more efficient computation for quantization in each computing node.

A parallel work by D. Alistarh *et al.* [29] presented QSGD that explores the trade-off between accuracy and gradient precision. The effectiveness of gradient quantization was justified and the convergence of QSGD was provably guaranteed. Compared to QSGD developed simultaneously, our *TernGrad* shares the same concept but advances in the following three aspects: (1) we prove the convergence from the perspective of statistic bound on gradients. The bound also explains why multiple quantization buckets are necessary in QSGD; (2) the bound is used to guide practices and inspires techniques of *layer-wise ternarizing* and *gradient clipping*; (3) *TernGrad* using only 3-level gradients achieves $0.92\%$ top-1 accuracy *improvement* for *AlexNet*, while $1.73\%$ top-1 accuracy *loss* is observed in QSGD with $4$ levels. The accuracy loss in QSGD can be eliminated by paying the cost of increasing the precision to 4 bits (16 levels) and beyond.

## 3 Problem Formulation and Our Approach

### 3.1 Problem Formulation and *TernGrad*

Figure 1 formulates the distributed training problem of synchronous SGD using data parallelism. At iteration $t$, a mini-batch of training samples are split and fed into multiple workers ($i \in \{1, ..., N\}$). Worker $i$ computes the gradients $\boldsymbol{g}_t^{(i)}$ of parameters *w.r.t.* its input samples $\boldsymbol{z}_t^{(i)}$. All gradients are

first synchronized and averaged at *parameter server*, and then sent back to update workers. Note that parameter server in most implementations [1][12] are used to preserve shared *parameters*, while here we utilize it in a slightly different way of maintaining shared *gradients*. In Figure 1, each worker keeps a copy of parameters locally. We name this technique as *parameter localization*. The parameter consistency among workers can be maintained by random initialization with an identical seed. *Parameter localization* changes the communication of parameters in floating-point form to the transfer of quantized gradients that require much lighter traffic. Note that our proposed *TernGrad* can be integrated with many settings like *Asynchronous SGD* [1][4], even though the scope of this paper only focuses on the distributed SGD in Figure 1.

**Algorithm 1** formulates the $t$-th iteration of *TernGrad* algorithm according to Figure 1. Most steps of *TernGrad* remain the same as traditional distributed training, except that gradients shall be quantized into ternary precision before sending to parameter server. More specific, $ternarize(\cdot)$ aims to reduce the communication volume of gradients. It randomly quantizes gradient $\boldsymbol{g}_t$ [2] to a ternary vector with values $\in \{-1, 0, +1\}$. Formally, with a random binary vector $\boldsymbol{b}_t$, $\boldsymbol{g}_t$ is ternarized as

$$\tilde{\boldsymbol{g}}_t = ternarize(\boldsymbol{g}_t) = s_t \cdot sign\left(\boldsymbol{g}_t\right) \circ \boldsymbol{b}_t, \tag{1}$$

where

$$s_t \triangleq max\left(abs\left(\boldsymbol{g}_t\right)\right) \triangleq ||\boldsymbol{g}_t||_\infty \tag{2}$$

is a *scaler*, *e.g. maximum norm*, that can shrink $\pm 1$ to a much smaller amplitude. $\circ$ is the Hadamard product. $sign(\cdot)$ and $abs(\cdot)$ respectively returns the sign and absolute value of each element. Giving a $\boldsymbol{g}_t$, each element of $\boldsymbol{b}_t$ independently follows the Bernoulli distribution

$$\begin{cases} P(b_{tk} = 1 \mid \boldsymbol{g}_t) = |g_{tk}|/s_t \\ P(b_{tk} = 0 \mid \boldsymbol{g}_t) = 1 - |g_{tk}|/s_t \end{cases}, \tag{3}$$

where $b_{tk}$ and $g_{tk}$ is the $k$-th element of $\boldsymbol{b}_t$ and $\boldsymbol{g}_t$, respectively. This *stochastic rounding*, instead of deterministic one, is chosen by both our study and QSGD [29], as stochastic rounding has an unbiased expectation and has been successfully studied for low-precision processing [20][30].

Theoretically, ternary gradients can at least reduce the *worker-to-server* traffic by a factor of $32/log_2(3) = 20.18\times$. Even using 2 bits to encode a ternary gradient, the reduction factor is still $16\times$. In this work, we compare *TernGrad* with 32-bit gradients, considering 32-bit is the default precision in modern deep learning frameworks. Although a lower-precision (*e.g.* 16-bit) may be enough in some scenarios, it will not undervalue *TernGrad*. As aforementioned, *parameter localization* reduces *server-to-worker* traffic by pulling quantized gradients from servers. However, summing up ternary values in $\sum_i \tilde{\boldsymbol{g}}_t^{(i)}$ will produce more possible levels and thereby the final averaged gradient $\overline{\boldsymbol{g}_t}$ is no longer ternary as shown in Figure 2(d). It emerges as a critical issue when workers use different scalers $s_t^{(i)}$. To minimize the number of levels, we propose a shared scaler

$$s_t = max(\{s_t^{(i)}\} : i = 1...N) \tag{4}$$

across all the workers. We name this technique as *scaler sharing*. The sharing process has a small overhead of transferring $2N$ floating scalars. By integrating *parameter localization* and *scaler sharing*, the maximum number of levels in $\overline{\boldsymbol{g}_t}$ decreases to $2N + 1$. As a result, the *server-to-worker* communication reduces by a factor of $32/log_2(1 + 2N)$, unless $N \geq 2^{30}$.

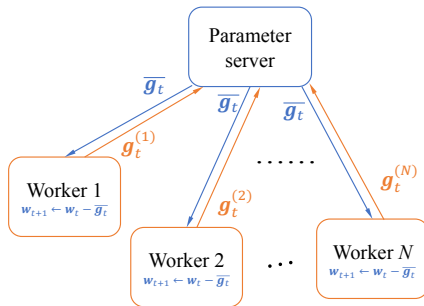

Figure 1: Distributed SGD with data parallelism.

**Algorithm 1** *TernGrad*: distributed SGD training using ternary gradients.

**Worker :** $i = 1, ..., N$

1    Input $\boldsymbol{z}_t^{(i)}$, a part of a mini-batch of training samples $\boldsymbol{z}_t$

2    Compute gradients $\boldsymbol{g}_t^{(i)}$ under $\boldsymbol{z}_t^{(i)}$

3    Ternarize gradients to $\tilde{\boldsymbol{g}}_t^{(i)} = ternarize(\boldsymbol{g}_t^{(i)})$

4    Push ternary $\tilde{\boldsymbol{g}}_t^{(i)}$ to the server

5    Pull averaged gradients $\overline{\boldsymbol{g}_t}$ from the server

6    Update parameters $\boldsymbol{w}_{t+1} \leftarrow \boldsymbol{w}_t - \eta \cdot \overline{\boldsymbol{g}_t}$

**Server :**

7    Average ternary gradients $\overline{\boldsymbol{g}_t} = \sum_i \tilde{\boldsymbol{g}}_t^{(i)}/N$

## 3.2 Convergence Analysis and Gradient Bound

We analyze the convergence of *TernGrad* in the framework of online learning systems. An online learning system adapts its parameter $\boldsymbol{w}$ to a sequence of observations to maximize performance. Each observation $\boldsymbol{z}$ is drawn from an unknown distribution, and a loss function $Q(\boldsymbol{z}, \boldsymbol{w})$ is used to measure the performance of current system with parameter $\boldsymbol{w}$ and input $\boldsymbol{z}$. The minimization target then is the loss expectation

$$C(\boldsymbol{w}) \triangleq \mathbf{E}\{Q(\boldsymbol{z}, \boldsymbol{w})\}. \tag{5}$$

In *General Online Gradient Algorithm* (GOGA) [33], parameter is updated at learning rate $\gamma_t$ as

$$\boldsymbol{w}_{t+1} = \boldsymbol{w}_t - \gamma_t \boldsymbol{g}_t = \boldsymbol{w}_t - \gamma_t \cdot \nabla_{\boldsymbol{w}} Q(\boldsymbol{z}_t, \boldsymbol{w}_t), \tag{6}$$

where

$$\boldsymbol{g} \triangleq \nabla_{\boldsymbol{w}} Q(\boldsymbol{z}, \boldsymbol{w}) \tag{7}$$

and the subscript $t$ denotes observing step $t$. In GOGA, $\mathbf{E}\{\boldsymbol{g}\}$ is the gradient of the minimization target in Eq. (5).

According to Eq. (1), the parameter in *TernGrad* is updated, such as

$$\boldsymbol{w}_{t+1} = \boldsymbol{w}_t - \gamma_t \left(s_t \cdot sign\left(\boldsymbol{g}_t\right) \circ \boldsymbol{b}_t\right), \tag{8}$$

where $s_t \triangleq ||\boldsymbol{g}_t||_\infty$ is a *random variable* depending on $\boldsymbol{z}_t$ and $\boldsymbol{w}_t$. As $\boldsymbol{g}_t$ is known for given $\boldsymbol{z}_t$ and $\boldsymbol{w}_t$, Eq. (3) is equivalent to

$$\begin{cases} P(b_{tk} = 1 \mid \boldsymbol{z}_t, \boldsymbol{w}_t) = |g_{tk}|/s_t \\ P(b_{tk} = 0 \mid \boldsymbol{z}_t, \boldsymbol{w}_t) = 1 - |g_{tk}|/s_t \end{cases}. \tag{9}$$

At any given $\boldsymbol{w}_t$, the expectation of ternary gradient satisfies

$$\mathbf{E}\{s_t \cdot sign\left(\boldsymbol{g}_t\right) \circ \boldsymbol{b}_t\} = \mathbf{E}\{s_t \cdot sign\left(\boldsymbol{g}_t\right) \circ \mathbf{E}\{\boldsymbol{b}_t | \boldsymbol{z}_t\}\} = \mathbf{E}\{\boldsymbol{g}_t\} = \nabla_{\boldsymbol{w}} C(\boldsymbol{w}_t), \tag{10}$$

which is an unbiased gradient of minimization target in Eq. (5).

The convergence analysis of *TernGrad* is adapted from the convergence proof of GOGA presented in [33]. We adopt two assumptions, which were used in analysis of the convergence of standard GOGA in [33]. Without explicit mention, vectors indicate column vectors here.

**Assumption 1.** *$C(\boldsymbol{w})$ has a single minimum $\boldsymbol{w}^*$ and gradient $-\nabla_{\boldsymbol{w}} C(\boldsymbol{w})$ always points to $\boldsymbol{w}^*$, i.e.,*

$$\forall \epsilon > 0, \inf_{||\boldsymbol{w}-\boldsymbol{w}^*||^2 > \epsilon} (\boldsymbol{w} - \boldsymbol{w}^*)^T \nabla_{\boldsymbol{w}} C(\boldsymbol{w}) > 0. \tag{11}$$

Convexity is a subset of Assumption 1, and we can easily find non-convex functions satisfying it.

**Assumption 2.** *Learning rate $\gamma_t$ is positive and constrained as*

$$\begin{cases} \sum_{t=0}^{+\infty} \gamma_t^2 < +\infty \\ \sum_{t=0}^{+\infty} \gamma_t = +\infty \end{cases}, \tag{12}$$

*which ensures $\gamma_t$ decreases neither very fast nor very slow respectively.*

We define the square of distance between current parameter $\boldsymbol{w}_t$ and the minimum $\boldsymbol{w}^*$ as

$$h_t \triangleq ||\boldsymbol{w}_t - \boldsymbol{w}^*||^2, \tag{13}$$

where $||\cdot||$ is $\ell_2$ norm. We also define the set of all random variables before step $t$ as

$$\boldsymbol{X}_t \triangleq (\boldsymbol{z}_{1...t-1}, \boldsymbol{b}_{1...t-1}). \tag{14}$$

Under Assumption 1 and Assumption 2, using Lyapunov process and Quasi-Martingales convergence theorem, L. Bottou [33] proved

**Lemma 1.** *If $\exists A, B > 0$ s.t.*

$$\mathbf{E}\left\{\left(h_{t+1} - \left(1 + \gamma_t^2 B\right) h_t\right) | \boldsymbol{X}_t\right\} \le -2\gamma_t (\boldsymbol{w}_t - \boldsymbol{w}^*)^T \nabla_{\boldsymbol{w}} C(\boldsymbol{w}_t) + \gamma_t^2 A, \tag{15}$$

*then $C(\boldsymbol{z}, \boldsymbol{w})$ converges **almost surely** toward minimum $\boldsymbol{w}^*$, i.e., $P\left(\lim_{t \to +\infty} \boldsymbol{w}_t = \boldsymbol{w}^*\right) = 1$.*

We further make an assumption on the gradient as

**Assumption 3** (Gradient Bound). *The gradient $\boldsymbol{g}$ is bounded as*

$$\mathbf{E}\left\{||\boldsymbol{g}||_{\infty} \cdot ||\boldsymbol{g}||_1\right\} \leq A + B\,||\boldsymbol{w} - \boldsymbol{w}^*||^2, \tag{16}$$

*where $A, B > 0$ and $||\cdot||_1$ is $\ell_1$ norm.*

With Assumption 3 and Lemma 1, we prove Theorem 1 ( in **Supplementary Material**):

**Theorem 1.** *When online learning systems update as*

$$\boldsymbol{w}_{t+1} = \boldsymbol{w}_t - \gamma_t\left(s_t \cdot sign\left(\boldsymbol{g}_t\right) \circ \boldsymbol{b}_t\right) \tag{17}$$

*using stochastic ternary gradients, they converge **almost surely** toward minimum $\boldsymbol{w}^*$, i.e., $P\left(\lim_{t\to+\infty}\boldsymbol{w}_t = \boldsymbol{w}^*\right) = 1$.*

Comparing with the gradient bound of standard GOGA [33]

$$\mathbf{E}\left\{||\boldsymbol{g}||^2\right\} \leq A + B\,||\boldsymbol{w} - \boldsymbol{w}^*||^2, \tag{18}$$

the bound in Assumption 3 is stronger because

$$||\boldsymbol{g}||_{\infty} \cdot ||\boldsymbol{g}||_1 \geq ||\boldsymbol{g}||^2. \tag{19}$$

We propose *layer-wise ternarizing* and *gradient clipping* to make two bounds closer, which shall be explained in Section 3.3. A side benefit of our work is that, by following the similar proof procedure, we can prove the convergence of GOGA when Gaussian noise $\mathcal{N}(0, \sigma^2)$ is added to gradients [34], under the gradient bound of

$$\mathbf{E}\left\{||\boldsymbol{g}||^2\right\} \leq A + B\,||\boldsymbol{w} - \boldsymbol{w}^*||^2 - \sigma^2. \tag{20}$$

Although the bound is also stronger, Gaussian noise encourages active exploration of parameter space and improves accuracy as was empirically studied in [34]. Similarly, the randomness of ternary gradients also encourages space exploration and improves accuracy for some models, as shall be presented in Section 4.

### 3.3 Feasibility Considerations

The gradient bound of *TernGrad* in Assumption 3 is stronger than the bound in standard GOGA. Pushing the two bounds closer can improve the convergence of *TernGrad*. In Assumption 3, $||\boldsymbol{g}||_{\infty}$ is the maximum absolute value of *all* the gradients in the DNN. So, in a large DNN, $||\boldsymbol{g}||_{\infty}$ could be relatively much larger than most gradients, implying that the bound in *TernGrad* becomes much stronger. Considering the situation, we propose *layer-wise ternarizing* and *gradient clipping* to reduce $||\boldsymbol{g}||_{\infty}$ and therefore shrink the gap between these two bounds.

*Layer-wise ternarizing* is proposed based on the observation that the range of gradients in each layer changes as gradients are back propagated. Instead of adopting a large global maximum scaler,

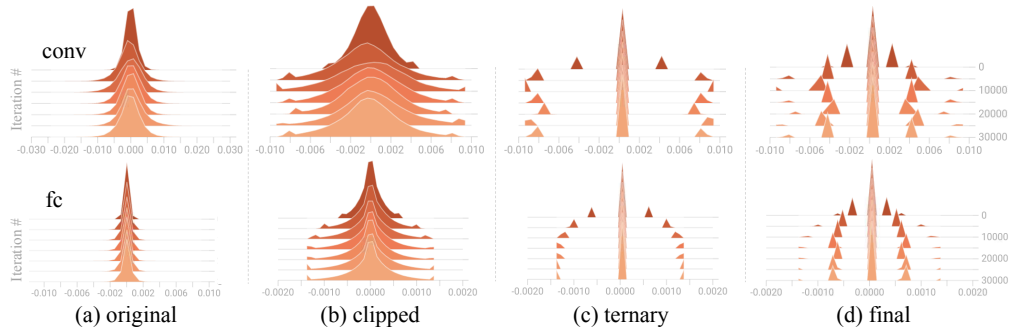

Figure 2: Histograms of (a) original floating gradients, (b) clipped gradients, (c) ternary gradients and (d) final averaged gradients. Visualization by TensorBoard. The DNN is *AlexNet* distributed on two workers, and vertical axis is the training iteration. As examples, top row visualizes the third convolutional layer and bottom one visualizes the first fully-connected layer.

we independently ternarize gradients in each layer using the layer-wise scalers. More specific, we separately ternarize the gradients of biases and weights by using Eq. (1), where $g_t$ could be the gradients of biases or weights in each layer. To approach the standard bound more closely, we can split gradients to more buckets and ternarize each bucket independently as D. Alistarh *et al.* [29] does. However, this will introduce more floating scalers and increase communication. When the size of bucket is one, it degenerates to floating gradients.

Layer-wise ternarizing can shrink the bound gap resulted from the dynamic ranges of the gradients across layers. However, the dynamic range within a layer still remains as a problem. We propose *gradient clipping*, which limits the magnitude of each gradient $g_i$ in $g$ as

$$f(g_i) = \begin{cases} g_i & |g_i| \leq c\sigma \\ sign(g_i) \cdot c\sigma & |g_i| > c\sigma \end{cases}, \tag{21}$$

where $\sigma$ is the standard derivation of gradients in $g$. In distributed training, gradient clipping is applied to every worker before ternarizing. $c$ is a hyper-parameter to select, but we cross validate it only once and use the constant in all our experiments. Specifically, we used a CNN [35] trained on CIFAR-10 by momentum SGD with staircase learning rate and obtained the optimal $c = 2.5$. Suppose the distribution of gradients is close to Gaussian distribution as shown in Figure 2(a), very few gradients can drop out of $[-2.5\sigma, 2.5\sigma]$. Clipping these gradients in Figure 2(b) can significantly reduce the scaler but slightly changes the length and direction of original $g$. Numerical analysis shows that *gradient clipping* with $c = 2.5$ only changes the length of $g$ by $1.0\% - 1.5\%$ and its direction by $2° - 3°$. In our experiments, $c = 2.5$ remains valid across multiple databases (MNIST, CIFAR-10 and ImageNet), various network structures (*LeNet*, *CifarNet*, *AlexNet*, *GoogLeNet*, *etc*) and training schemes (momentum, vanilla SGD, adam, *etc*).

The effectiveness of *layer-wise ternarizing* and *gradient clipping* can also be explained as follows. When the scalar $s_t$ in Eq. (1) and Eq. (3) is very large, most gradients have a high possibility to be ternarized to zeros, leaving only a few gradients to large-magnitude values. The scenario raises a severe parameter update pattern: most parameters keep unchanged while others likely overshoot. This will introduce large training variance. Our experiments on *AlexNet* show that by applying both *layer-wise ternarizing* and *gradient clipping* techniques, *TernGrad* can converge to the same accuracy as standard SGD. Removing any of the two techniques can result in accuracy degradation, *e.g.*, $3\%$ top-1 accuracy loss without applying *gradient clipping* as we shall show in Table 2.

## 4 Experiments

We first investigate the convergence of *TernGrad* under various training schemes on relatively small databases and show the results in Section 4.1. Then the scalability of *TernGrad* to large-scale distributed deep learning is explored and discussed in Section 4.2. The experiments are performed by TensorFlow[2]. We maintain the exponential moving average of parameters by employing an exponential decay of $0.9999$ [15]. The accuracy is evaluated by the final averaged parameters. This gives slightly better accuracy in our experiments. For fair comparison, in each pair of comparative experiments using either floating or ternary gradients, all the other training hyper-parameters are the same unless differences are explicitly pointed out. In experiments, when SGD with momentum is adopted, momentum value of $0.9$ is used. When polynomial decay is applied to decay the *learning rate* (LR), the power of $0.5$ is used to decay LR from the base LR to zero.

### 4.1 Integrating with Various Training Schemes

We study the convergence of *TernGrad* using *LeNet* on MNIST and a ConvNet [35] (named as *CifarNet*) on CIFAR-10. *LeNet* is trained without data augmentation. While training *CifarNet*, images

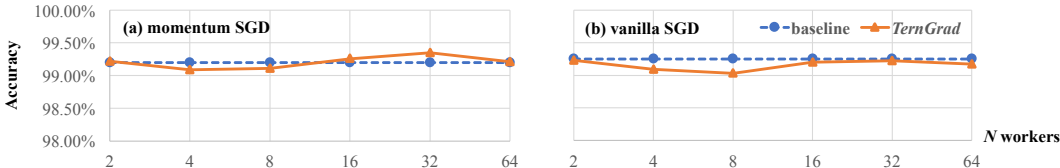

Figure 3: Accuracy vs. worker number for baseline and *TernGrad*, trained with (a) momentum SGD or (b) vanilla SGD. In all experiments, total mini-batch size is $64$ and maximum iteration is $10K$.

Table 1: Results of *TernGrad* on *CifarNet*.

| SGD | base LR | total mini-batch size | iterations | gradients | workers | accuracy |
|------|---------|----------------------|-----------|-----------|---------|----------|
| Adam | 0.0002 | 128 | 300K | floating | 2 | 86.56% |
| | | | | *TernGrad* | 2 | 85.64% (-0.92%) |
| Adam | 0.0002 | 2048 | 18.75K | floating | 16 | 83.19% |
| | | | | *TernGrad* | 16 | 82.80% (-0.39%) |

are randomly cropped to $24 \times 24$ images and mirrored. Brightness and contrast are also randomly adjusted. During the testing of *CifarNet*, only center crop is used. Our experiments cover the scope of SGD optimizers over vanilla SGD, SGD with momentum [36] and Adam [37].

Figure 3 shows the results of *LeNet*. All are trained using polynomial LR decay with weight decay of 0.0005. The base learning rates of momentum SGD and vanilla SGD are 0.01 and 0.1, respectively. Given the total mini-batch size $M$ and the worker number $N$, the mini-batch size per worker is $M/N$. Without explicit mention, mini-batch size refers to the total mini-batch size in this work. Figure 3 shows that *TernGrad* can converge to the similar accuracy within the same iterations, using momentum SGD or vanilla SGD. The maximum accuracy gain is $0.15\%$ and the maximum accuracy loss is $0.22\%$. Very importantly, the communication time per iteration can be reduced. The figure also shows that *TernGrad* generalizes well to distributed training with large $N$. No degradation is observed even for $N = 64$, which indicates one training sample per iteration per worker.

Table 1 summarizes the results of *CifarNet*, where all trainings terminate after the same epochs. Adam SGD is used for training. Instead of keeping total mini-batch size unchanged, we maintain the mini-batch size per worker. Therefore, the total mini-batch size linearly increases as the number of workers grows. Though the base learning rate of $0.0002$ seems small, it can achieve better accuracy than larger ones like $0.001$ for baseline. In each pair of experiments, *TernGrad* can converge to the accuracy level with less than 1% degradation. The accuracy degrades under a large mini-batch size in both baseline and *TernGrad*. This is because parameters are updated less frequently and large-batch training tends to converge to poorer sharp minima [38]. However, the noise inherent in *TernGrad* can help converge to better flat minimizers [38], which could explain the smaller accuracy gap between the baseline and *TernGrad* when the mini-batch size is 2048. In our experiments of *AlexNet* in Section 4.2, *TernGrad* even improves the accuracy in the large-batch scenario. This attribute is beneficial for distributed training as a large mini-batch size is usually required.

## 4.2 Scaling to Large-scale Deep Learning

We also evaluate *TernGrad* by *AlexNet* and *GoogLeNet* trained on ImageNet. It is more challenging to apply *TernGrad* to large-scale DNNs. It may result in some accuracy loss when simply replacing the floating gradients with ternary gradients while keeping other hyper-parameters unchanged. However, we are able to train large-scale DNNs by *TernGrad* successfully after making some or all of the following changes: (1) decreasing dropout ratio to keep more neurons; (2) using smaller weight decay; and (3) disabling ternarizing in the last classification layer. Dropout can regularize DNNs by adding randomness, while *TernGrad* also introduces randomness. Thus, dropping fewer neurons helps avoid over-randomness. Similarly, as the randomness of *TernGrad* introduces regularization, smaller weight decay may be adopted. We suggest not to apply ternarizing to the last layer, considering that the one-hot encoding of labels generates a skew distribution of gradients and the symmetric ternary encoding $\{-1, 0, 1\}$ is not optimal for such a skew distribution. Though asymmetric ternary levels could be an option, we decide to stick to floating gradients in the last layer for simplicity. The overhead of communicating these floating gradients is small, as the last layer occupies only a small percentage of total parameters, like 6.7% in *AlexNet* and 3.99% in *ResNet-152* [39].

All DNNs are trained by momentum SGD with Batch Normalization [40] on convolutional layers. *AlexNet* is trained by the hyper-parameters and data augmentation depicted in Caffe. *GoogLeNet* is trained by polynomial LR decay and data augmentation in [41]. Our implementation of *GoogLeNet* does not utilize any auxiliary classifiers, that is, the loss from the last softmax layer is the total loss. More training hyper-parameters are reported in corresponding tables and published source code. Validation accuracy is evaluated using only the central crops of images.

The results of *AlexNet* are shown in Table 2. Mini-batch size per worker is fixed to 128. For fast development, all DNNs are trained through the same epochs of images. In this setting, when there are

Table 2: Accuracy comparison for *AlexNet*.

| base LR | mini-batch size | workers | iterations | gradients | weight decay | DR[†] | top-1 | top-5 |
|---------|-----------------|---------|------------|-----------|--------------|-------|-------|-------|
| 0.01 | 256 | 2 | 370K | floating | 0.0005 | 0.5 | 57.33% | 80.56% |
| | | | | *TernGrad* | 0.0005 | 0.2 | 57.61% | 80.47% |
| | | | | *TernGrad*-noclip [‡] | 0.0005 | 0.2 | 54.63% | 78.16% |
| 0.02 | 512 | 4 | 185K | floating | 0.0005 | 0.5 | 57.32% | 80.73% |
| | | | | *TernGrad* | 0.0005 | 0.2 | 57.28% | 80.23% |
| 0.04 | 1024 | 8 | 92.5K | floating | 0.0005 | 0.5 | **56.62%** | 80.28% |
| | | | | *TernGrad* | 0.0005 | 0.2 | **57.54%** | 80.25% |

[†] DR: dropout ratio, the ratio of dropped neurons. [‡] *TernGrad* without gradient clipping.

Table 3: Accuracy comparison for *GoogLeNet*.

| base LR | mini-batch size | workers | iterations | gradients | weight decay | DR | top-5 |
|---------|-----------------|---------|------------|-----------|--------------|-----|-------|
| 0.04 | 128 | 2 | 600K | floating | 4e-5 | 0.2 | 88.30% |
| | | | | *TernGrad* | 1e-5 | 0.08 | 86.77% |
| 0.08 | 256 | 4 | 300K | floating | 4e-5 | 0.2 | 87.82% |
| | | | | *TernGrad* | 1e-5 | 0.08 | 85.96% |
| 0.10 | 512 | 8 | 300K | floating | 4e-5 | 0.2 | 89.00% |
| | | | | *TernGrad* | 2e-5 | 0.08 | 86.47% |

more workers, the number of iterations becomes smaller and parameters are less frequently updated. To overcome this problem, we increase the learning rate for large-batch scenario [10]. Using this scheme, SGD with floating gradients successfully trains *AlexNet* to similar accuracy, for mini-batch size of 256 and 512. However, when mini-batch size is 1024, the top-1 accuracy drops 0.71% for the same reason as we point out in Section 4.1.

*TernGrad* converges to approximate accuracy levels regardless of mini-batch size. Notably, it improves the top-1 accuracy by 0.92% when mini-batch size is 1024, because its inherent randomness encourages to escape from poorer sharp minima [34][38]. Figure 4 plots training details vs. iteration when mini-batch size is 512. Figure 4(a) shows that the convergence curve of *TernGrad* matches well with the baseline's, demonstrating the effectiveness of *TernGrad*. The training efficiency can be further improved by reducing communication time as shall be discussed in Section 5. The training data loss in Figure 4(b) shows that *TernGrad* converges to a slightly lower level, which further proves the capability of *TernGrad* to minimize the target function even with ternary gradients. A smaller dropout ratio in *TernGrad* can be another reason of the lower loss. Figure 4(c) simply illustrate that on average 71.32% gradients of a fully-connected layer (fc6) are ternarized to zeros.

Finally, we summarize the results of *GoogLeNet* in Table 3. On average, the accuracy loss is less than 2%. In *TernGrad*, we adopted all that hyper-parameters (except dropout ratio and weight decay) that are well tuned for the baseline [42]. Tuning these hyper-parameters specifically for *TernGrad* could further optimize *TernGrad* and obtain higher accuracy.

## 5 Performance Model and Discussion

Our proposed *TernGrad* requires only three numerical levels $\{-1, 0, 1\}$, which can aggressively reduce the communication time. Moreover, our experiments in Section 4 demonstrate that within *the*

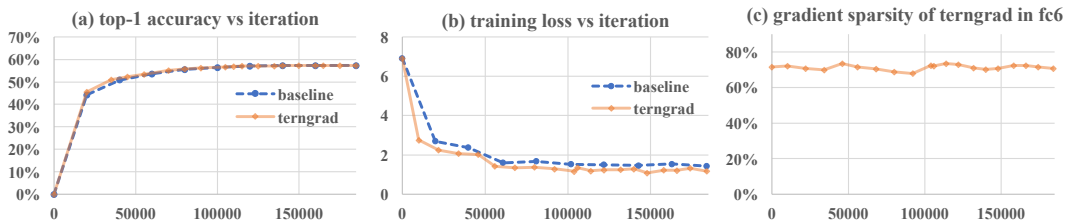

Figure 4: *AlexNet* trained on 4 workers with mini-batch size 512: (a) top-1 validation accuracy, (b) training data loss and (c) sparsity of gradients in first fully-connected layer (fc6) vs. iteration.

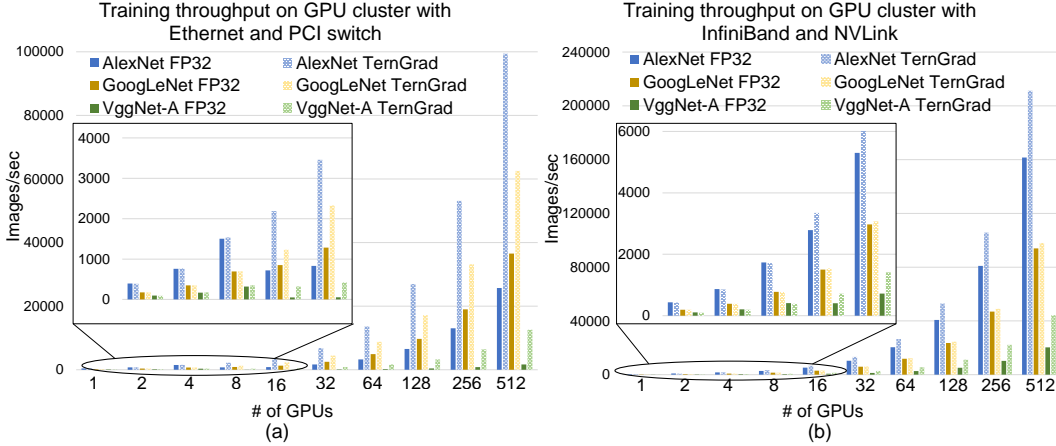

Figure 5: Training throughput on two different GPUs clusters: (a) 128-node GPU cluster with 1Gbps Ethernet, each node has 4 NVIDIA GTX 1080 GPUs and one PCI switch; (b) 128-node GPU cluster with 100 Gbps InfiniBand network connections, each node has 4 NVIDIA Tesla P100 GPUs connected via NVLink. Mini-batch size per GPU of *AlexNet*, *GoogLeNet* and *VggNet-A* is 128, 64 and 32, respectively

*same iterations*, *TernGrad* can converge to *approximately the same accuracy* as its corresponding baseline. Consequently, a dramatical throughput improvement on the distributed DNN training is expected. Due to the resource and time constraint, unfortunately, we aren't able to perform the training of more DNN models like *VggNet-A* [43] and distributed training beyond 8 workers. We plan to continue the experiments in our future work. We opt for using a performance model to conduct the scalability analysis of DNN models when utilizing up to 512 GPUs, with and without applying *TernGrad*. Three neural network models—*AlexNet*, *GoogLeNet* and *VggNet-A*—are investigated. In discussions of performance model, *performance* refers to training speed. Here, we extend the performance model that was initially developed for CPU-based deep learning systems [44] to estimate the performance of distributed GPUs/machines. The key idea is combining the lightweight profiling on single machine with analytical modeling for accurate performance estimation. In the interest of space, please refer to **Supplementary Material** for details of the performance model.

Figure 5 presents the training throughput on two different GPUs clusters. Our results show that *TernGrad* effectively increases the training throughput for the three DNNs. The speedup depends on the communication-to-computation ratio of the DNN, the number of GPUs, and the communication bandwidth. DNNs with larger communication-to-computation ratios (*e.g. AlexNet* and *VggNet-A*) can benefit more from *TernGrad* than those with smaller ratios (*e.g.*, *GoogLeNet*). Even on a very high-end HPC system with InfiniBand and NVLink, *TernGrad* is still able to double the training speed of *VggNet-A* on 128 nodes as shown in Figure 5(b). Moreover, the *TernGrad* becomes more efficient when the bandwidth becomes smaller, such as 1Gbps Ethernet and PCI switch in Figure 5(a) where *TernGrad* can have $3.04\times$ training speedup for *AlexNet* on 8 GPUs.

### Acknowledgments

This work was supported in part by NSF CCF-1744082 and DOE SC0017030. Any opinions, findings, conclusions or recommendations expressed in this material are those of the authors and do not necessarily reflect the views of NSF, DOE, or their contractors. Thanks Ali Taylan Cemgil at Bogazici University for valuable suggestions on this work.

## Footnotes

[1]https://github.com/wenwei202/terngrad

[2]Here, the superscript of $\boldsymbol{g}_t$ is omitted for simplicity.

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
