[Supplementary Material]

# Supplementary Material – TernGrad: Ternary Gradients to Reduce Communication in Distributed Deep Learning

**Wei Wen[1], Cong Xu[2], Feng Yan[3], Chunpeng Wu[1], Yandan Wang[4], Yiran Chen[1], Hai Li[1]**

[1]Duke University, [2]Hewlett Packard Labs, [3]University of Nevada – Reno, [4]University of Pittsburgh
[1]{wei.wen, chunpeng.wu, yiran.chen, hai.li}@duke.edu
[2]cong.xu@hpe.com, [3]fyan@unr.edu, [4]yaw46@pitt.edu

## Abstract

This supplementary material provides the proof of the convergence of *TernGrad*, and details of our performance model.

## 1  Convergence Analysis of *TernGrad*

**Theorem 1.** *When online learning systems update as* $\boldsymbol{w}_{t+1} = \boldsymbol{w}_t - \gamma_t \left( s_t \cdot sign\left(\boldsymbol{g}_t\right) \circ \boldsymbol{b}_t \right)$ *using stochastic ternary gradients, they converge **almost surely** toward minimum* $\boldsymbol{w}^*$*, i.e.,* $P\left(\lim_{t \to +\infty} \boldsymbol{w}_t = \boldsymbol{w}^*\right) = 1$.

*Proof.*

$$h_{t+1} - h_t = -2\gamma_t(\boldsymbol{w}_t - \boldsymbol{w}^*)^T \left(s_t \cdot sign\left(\boldsymbol{g}_t\right) \circ \boldsymbol{b}_t\right) + \gamma_t^2 \left|\left|s_t \cdot sign\left(\boldsymbol{g}_t\right) \circ \boldsymbol{b}_t\right|\right|^2. \tag{1}$$

We have

$$\begin{aligned}
\mathbf{E}\left\{\left(h_{t+1} - h_t\right)|\boldsymbol{X}_t\right\} &= -2\gamma_t(\boldsymbol{w}_t - \boldsymbol{w}^*)^T \mathbf{E}\left\{\left(s_t \cdot sign\left(\boldsymbol{g}_t\right) \circ \boldsymbol{b}_t\right)|\boldsymbol{X}_t\right\} \\
&\quad + \gamma_t^2 \mathbf{E}\left\{\left.\left|\left|s_t \cdot sign\left(\boldsymbol{g}_t\right) \circ \boldsymbol{b}_t\right|\right|^2\right|\boldsymbol{X}_t\right\}.
\end{aligned} \tag{2}$$

Eq. (2) satisfies based on the fact that $\gamma_t$ is deterministic, and $\boldsymbol{w}_t$ is also deterministic given $\boldsymbol{X}_t$. According to $\mathbf{E}\left\{s_t \cdot sign\left(\boldsymbol{g}_t\right) \circ \boldsymbol{b}_t\right\} = \nabla_{\boldsymbol{w}} C(\boldsymbol{w}_t)$,

$$\begin{aligned}
&\mathbf{E}\left\{\left(h_{t+1} - h_t\right)|\boldsymbol{X}_t\right\} + 2\gamma_t \cdot (\boldsymbol{w}_t - \boldsymbol{w}^*)^T \cdot \nabla_{\boldsymbol{w}} C(\boldsymbol{w}_t) \\
&= \gamma_t^2 \cdot \mathbf{E}\left\{\left.\left|\left|s_t \cdot sign\left(\boldsymbol{g}_t\right) \circ \boldsymbol{b}_t\right|\right|^2\right|\boldsymbol{X}_t\right\} \\
&= \gamma_t^2 \cdot \mathbf{E}\left\{\left. s_t^2 \left|\left|\boldsymbol{b}_t\right|\right|^2 \right|\boldsymbol{w}_t\right\} = \gamma_t^2 \cdot \mathbf{E}\left\{\left. s_t^2 \cdot \mathbf{E}\left\{\left.\left|\left|\boldsymbol{b}_t\right|\right|^2\right|\boldsymbol{z}_t, \boldsymbol{w}_t\right\}\right|\boldsymbol{w}_t\right\} \\
&= \gamma_t^2 \cdot \mathbf{E}\left\{\left. s_t^2 \cdot \sum_k \mathbf{E}\left\{\left. b_{tk}^2\right|\boldsymbol{z}_t, \boldsymbol{w}_t\right\}\right|\boldsymbol{w}_t\right\}
\end{aligned} \tag{3}$$

Based on the Bernoulli distribution of $b_{tk}$ and Assumption 3, we further have

$$\begin{aligned}
&\mathbf{E}\left\{\left(h_{t+1} - h_t\right)|\boldsymbol{X}_t\right\} + 2\gamma_t \cdot (\boldsymbol{w}_t - \boldsymbol{w}^*)^T \cdot \nabla_{\boldsymbol{w}} C(\boldsymbol{w}_t) \\
&= \gamma_t^2 \cdot \mathbf{E}\left\{s_t \left|\left|\boldsymbol{g}_t\right|\right|_1\right\} = \gamma_t^2 \cdot \mathbf{E}\left\{max\left(abs\left(\boldsymbol{g}_t\right)\right) \cdot \left|\left|\boldsymbol{g}_t\right|\right|_1\right\} \\
&\leq A\gamma_t^2 + B\gamma_t^2 \left|\left|\boldsymbol{w}_t - \boldsymbol{w}^*\right|\right|^2 = A\gamma_t^2 + B\gamma_t^2 h_t.
\end{aligned} \tag{4}$$

That is

$$\mathbf{E}\left\{\left(h_{t+1} - \left(1 + \gamma_t^2 B\right) h_t\right)|\boldsymbol{X}_t\right\} \leq -2\gamma_t(\boldsymbol{w}_t - \boldsymbol{w}^*)^T \nabla_{\boldsymbol{w}} C(\boldsymbol{w}_t) + \gamma_t^2 A, \tag{5}$$

which satisfies the condition of Lemma 1 and proves Theorem 1. The proof can be extended to mini-batch SGD by treating $\boldsymbol{z}$ as a mini-batch of observations instead of one observation.  $\square$

## 2 Performance Model

As mentioned in the main context of our paper, the performance model was developed based on the one initially proposed for CPU-based deep learning systems [1]. We extended it to model GPU-based deep learning systems in this work. Lightweight profiling is used in the model. We ran all performance tests with distributed TensorFlow on a cluster of 4 machines, each of which has 4 GTX 1080 GPUs and one Mellanox MT27520 InfiniBand network card. Our performance model was successfully validated against the measured results by the server cluster we have.

There are two scaling schemes for distributed training with data parallelism: a) *strong scaling* that spreads the same size problem across multiple workers, and b) *weak scaling* that keeps the size per worker constant when the number of workers increases [2]. Our performance model supports both scaling models.

We start with strong scaling to illustrate our performance model. According to the definition of strong scaling, here the same size problem is corresponding to the same mini-batch size. In other words, the more workers, the less training samples per worker. Intuitively, more workers bring more computing resources, meanwhile inducing higher communication overhead. The goal is to estimate the throughput of a system that uses $j$ machines with $i$ GPUs per machine and mini-batch size of $K$[1]. Note the total number of workers equals to the total number of GPUs on all machines, i.e., $N = i * j$. We need to distinguish workers within a machine and across machines due to their different communication patterns. Next, we illustrate how to accurately model the impacts in communication and computation to capture both the benefits and overheads.

**Communication.** For GPUs within a machine, first, the gradient $g$ computed at each GPU needs to be accumulated together. Here we assume all-reduce communication model, that is, each GPU communicates with its neighbor until all gradient $g$ is accumulated into a single GPU. The communication complexity for $i$ GPUs is $log_2 i$. The GPU with accumulated gradient then sends the accumulated gradient to CPU for further processing. Note for each communication (either GPU-to-GPU or GPU-to-CPU), the communication data size is the same, i.e., $|g|$. Assume that within a machine, the communication bandwidth between GPUs is $C_{gwd}$[2] and the communication bandwidth between CPU and GPU is $C_{cwd}$, then the communication overhead within a machine can be computed as $\frac{|g|}{C_{gwd}} * log_2 i + \frac{|g|}{C_{cwd}}$. We successfully used NCCL benchmark to validate our model. For communication between machines, we also assume all-reduce communication model, so the communication time between machines are: $(C_{ncost} + \frac{|g|}{C_{nwd}}) * log_2 j$, where $C_{ncost}$ is the network latency and $C_{nwd}$ is the network bandwidth. So the total communication time is $T_{comm}(i, j, K, |g|) = \frac{|g|}{C_{gwd}} * log_2 i + \frac{|g|}{C_{cwd}} + (C_{ncost} + \frac{|g|}{C_{nwd}}) * log_2 j$. We successfully used OSU Allreduce benchmark to validate this model.

**Computation.** To estimate computation time, we rely on profiling the time for training a mini-batch of totally $K$ images on a machine with a single CPU and a single GPU. We define this profiled time as $T(1, 1, K, |g|)$. In strong scaling, each work only trains $\frac{K}{N}$ samples, so the total computation time is $T_{comp}(i, j, K, |g|) = (T(1, 1, K, |g|) - \frac{|g|}{C_{cwd}}) * \frac{1}{N}$, where $\frac{|g|}{C_{cwd}}$ is the communication time (between GPU and CPU) included in when we profile $T(1, 1, K, |g|)$.

Therefore, the time to train a mini-batch of $K$ samples is:

$$
\begin{aligned}
T_{strong}(i, j, K, |g|) &= T_{comp}(i, j, K, |g|) + T_{comm}(i, j, K, |g|) \\
&= (T(1, 1, K, |g|) - \frac{|g|}{C_{cwd}}) * \frac{1}{N} \\
&+ \frac{|g|}{C_{gwd}} * log_2 i + \frac{|g|}{C_{cwd}} + (C_{ncost} + \frac{|g|}{C_{nwd}}) * log_2 j.
\end{aligned}
\tag{6}
$$

The throughput of strong scaling is:

$$
Tput_{strong}(i, j, K, |g|) = \frac{K}{T_{strong}(i, j, K, |g|)}.
\tag{7}
$$

For weak scaling, the difference is that each worker always trains $K$ samples. So the mini-batch size becomes $N * K$. In the interest of space, we do not present the detailed reasoning here. Basically, it follows the same logic for developing the performance model of strong scaling. We can compute the time to train a mini-batch of $N * K$ samples as follows:

$$
\begin{aligned}
T_{weak}(i, j, K, |\boldsymbol{g}|) &= T_{comp}(i, j, K, |\boldsymbol{g}|) + T_{comm}(i, j, K, |\boldsymbol{g}|) \\
&= T(1, 1, K, |\boldsymbol{g}|) - \frac{|\boldsymbol{g}|}{C_{cwd}} + \frac{|\boldsymbol{g}|}{C_{gwd}} * log_2 i + \frac{|\boldsymbol{g}|}{C_{cwd}} + (C_{ncost} + \frac{|\boldsymbol{g}|}{C_{nwd}}) * log_2 j \\
&= T(1, 1, K, |\boldsymbol{g}|) + \frac{|\boldsymbol{g}|}{C_{gwd}} * log_2 i + (C_{ncost} + \frac{|\boldsymbol{g}|}{C_{nwd}}) * log_2 j.
\end{aligned}
\tag{8}
$$

So the throughput of weak scaling is:

$$
Tput_{weak}(i, j, K, |\boldsymbol{g}|) = \frac{N * K}{T_{weak}(i, j, K, |\boldsymbol{g}|)}.
\tag{9}
$$

## Footnotes

[1]For ease of the discussion, we assume symmetric system architecture. The performance model can be easily extended to support heterogeneous system architecture.

[2]For ease of the discussion, we assume GPU-to-GPU communication has *Dedicated Bandwidth*.