[Reviews · NeurIPS 2017]

Reviewer 1



The paper proposes a mechanism to reduce the traffic of sending gradients from the workers to the server, by heavily quantizing them. Unlike previous work, they quantize the gradients into the three levels as the name of the paper implies. They show that this allows DNN training from scratch and provides convergence guarantees. Thus, this is a significant improvement over the 1-bit quantization case. Suggestion: The authors should elaborate the choices made in layer-wise ternarizing in the experiments reported. An analysis related to Line 168-169 would also be useful to understand the trade-offs.

Reviewer 2



Basic idea is to speed-up distributed training by quantizing gradients to 3 values. Parameter Server syncs gradients, works keep local copy of parameters now and scalar is shared among workers. Paper is well written and easy to follow. Experimental results are strong and convincing. Good paper, overall. Line 106: You compare this to 32bit floats, but for many setups, quantizing to 16bit is reasonably safe and training converges to same accuracy than when using floats for the gradient. If you use then 4 bit for your gradient, you get a factor of 4 communication reduction when sending to the gradient server.

Reviewer 3



The authors tried to reduce communication cost in distributed deep learning. They proposed a new method called Ternary Gradients to reduce the overhead of gradient synchronization. The technique is correct and the authors made some experiments to evaluate their method. Overall, the method is novel and interesting and may have significant impact on large scale deep learning. Some comments below for the authors to improve their paper. 1. Some notations and words should be made clearer. For example, s is defined as the max value of abs(g), but the domain of the max operation fails to show up in the equation. 2. The author mentioned that they tried to narrow the gap between the two bounds in the paper for better convergence, but not mentioned for better convergence rate or better convergence probability. And it’s a pity that there’s no theoretical analysis of the convergence rate. 3. One big drawback is that the author made too strong assumption, i.e. the cost function has only one single minimum, which means the objective function is convex. It’s rare in practice. To my best knowledge, there’re many efficient algorithms for convex optimization in distributed systems. Therefore, the proposed method will not have big impacts. 4. The author should use more metrics in the experiments instead of just accuracy.